# Phage SPO1 Protein Gp49 Is a Novel RNA Binding Protein That Is Involved in Host Iron Metabolism

**DOI:** 10.3390/ijms241814318

**Published:** 2023-09-20

**Authors:** Yanan Yang, Zhenyue Hu, Yue Kang, Juanjuan Gao, Huan Chen, Hui Liu, Yawen Wang, Bing Liu

**Affiliations:** 1BioBank, The First Affiliated Hospital of Xi’an Jiaotong University, Yanta District, Xi’an 710061, China; yyn6979@stu.xjtu.edu.cn (Y.Y.); huzhenyue@xjtu.edu.cn (Z.H.); 3120115190@stu.xjtu.edu.cn (Y.K.); juan_gao2018@xjtu.edu.cn (J.G.); 3120115171@stu.xjtu.edu.cn (H.C.); liuh2018@xjtu.edu.cn (H.L.); 2Centre for Biobank and Advanced Medical Research of Shaanxi Province, The First Affiliated Hospital of Xi’an Jiaotong University, Xi’an 710061, China

**Keywords:** SPO1, Gp49, RNA binding protein, iron metabolism, Bacteriophages

## Abstract

*Bacillus subtilis* is a model organism for studying Gram-positive bacteria and serves as a cell factory in the industry for enzyme and chemical production. Additionally, it functions as a probiotic in the gastrointestinal tract, modulating the gut microbiota. Its lytic phage SPO1 is also the most studied phage among the genus *Okubovrius*, including Bacillus phage SPO1 and Camphawk. One of the notable features of SPO1 is the existence of a “host-takeover module”, a cluster of 24 genes which occupies most of the terminal redundancy. Some of the gene products from the module have been characterized, revealing their ability to disrupt host metabolism by inhibiting DNA replication, RNA transcription, cell division, and glycolysis. However, many of the gene products which share limited similarity to known proteins remain under researched. In this study, we highlight the involvement of Gp49, a gene product from the module, in host RNA binding and heme metabolism—no observation has been reported in other phages. Gp49 folds into a structure that does not resemble any protein in the database and has a new putative RNA binding motif. The transcriptome study reveals that Gp49 primarily upregulates host heme synthesis which captures cytosolic iron to facilitate phage development.

## 1. Introduction

Bacteriophage (phage) SPO1 is a lytic phage of *Bacillus subtilis* (*B. subtilis*) [1], which has been intensively studied and has provided conceptual advancements for phage biology [2]. Besides the similarity amongst Bacillus phages, some other known phages infecting Gram-positive bacteria have also a significant level of similarity to SPO1, including *Staphylococcus* phages K, G1 and Twort, *Listeria* phage P100, and *Lactobacillus* phage [3]. SPO1 has been the most studied among these phages and is well-known because of the extensive works as a model organism for many pioneering studies in bacterial and phage biology. These include the programmed sequential activation of phage gene expression via sigma factors and the formation of terminally redundant concatemers in its genome encoding proteins that are essential for host-takeover [2]. In particular, the 13,185 bp terminal redundancy makes up 10% of the 209 gene SPO1 genome sequence. Interestingly, around 60% of its 204 protein gene products has no similarity to any known protein. This high proportion of novel genes indicates the necessity to annotate SPO1 genome further for phage biology and protein science. Meanwhile, the thymine of SPO1 DNA is completely replaced by hydroxymethyluracil (hmUra), facilitating the specific recognition of SPO1 DNA binding protein [4]. For example, TF01 of SPO1 binds with high affinity to specific sites in the SPO1 chromosome but not the *B. subtilis* DNA [5].

The host-takeover module (HTM) in the terminal redundancy contains 24 genes (Figure 1A), all with low similarities to other characterized proteins. Recent advances in annotating these gene products revealed several interesting host hijacking strategies employed by the phage. For example, Gp44 is a non-specific transcription inhibitor which contains a DNA binding motif that binds to DNA and a DNA mimic domain that traps RNA polymerase (RNAP) [6]. Gp46 is a cross-species HU inhibitor that causes filamentous morphology and nucleoid deformation in *B. subtilis* [7]. PEIP (also known as Gp60) has been shown to be a potent glycolysis inhibitor that dissembles the octameric enolase that results in growth attenuation and disruption of the cell wall [8]. Other studies also suggest that Gp53 is a host ClpCP modifier [9] and Gp56 inhibits cell division by interacting with FtsL [10]. These studies suggest that the low sequence homologies in these gene products translate into novel functions or host takeover strategies that are rarely seen in other phage proteins.

Iron is essential for virtually all organisms, including bacteria and phages. Ferrous iron can be imported across the cell membrane by iron transporter FeoB, and stored in iron-containing proteins like bacterioferritin that acts as an internal iron reservoir when external iron is diminished [11]. The heme-containing bacterioferritins are only found in eubacteria and exist in a complex containing 24 subunits. This iron storage protein takes up iron in the soluble ferrous form but deposits it in its central cavity in the oxidized form, with a capacity of 2000–3000 iron atoms per complex [12]. However, over accumulation of iron poses an oxidative burden to bacteria; the toxicity for iron is sensed and the concentration is tightly regulated by adjusting the acquisition, storage, and efflux of iron in bacteria [13]. The bacteriophages have also been shown to subjected to inactivation by ferrous iron oxidation [14]. Albeit, the response from phages has not been described—the linear relationship between phage inactivation (log) and iron concentration.

Here, we report Gp49 of the HTM as a transcription regulator that promotes iron acquisition. Gp49 folds into a structure that does not resemble other published proteins and contains a putative RNA binding motif. Its interaction with RNA results in a clear difference in host transcriptome, i.e., upregulation of mRNA involved in bacterial bacterioferritin (heme) synthesis and iron metabolism. Thus, our study connects bacterial iron acquisition to the response from phage, and reveals a potential strategy employed by SPO1 to detoxicate iron from the environment or bacterial defense by upregulating bacterioferritin synthesis to acquire an excess amount of iron.

## 2. Results

### 2.1. Gp49 Is Only Conserved in Bacillus Phages and Might Be Involved in Host Nucleic Acid Metabolism

As the function of Gp49 was not clear, we first attempted to find clues from its adjacent genes in the HTM (Figure 1A). Unfortunately, little is known for the other gene—*gp48* in operon 4. However, gene products from adjacent operons shed light on its function. Among the neighboring genes whose functions have been annotated, gene product 44 is a non-specific RNAP binding protein, that is able to shut down RNA synthesis for both *E. coli* and *B. subtilis* protein [6]. In the neighboring operon 3, Gp46 is a host HU protein inhibitor whose overexpression causes the loss of nucleoid structure [7]. On the other side of *gp49*, products of gene 50 and 51 were considered to bind to host RNAP for regulating host phage gene expression while shutting off host RNA synthesis [15]. As all neighboring genes with known functions are involved in manipulating host nucleic acid metabolism, Gp49 might also have a similar function as seen in previously described phage gene clusters [16]. However, the homologue search using BLAST did not reveal any additional clue, as Gp49 is only found to be conserved in *Okubovirus* genus members that infects *B. subtilis* (Figure 1B). Thus, the potential role of Gp49 in host nucleic acid metabolism requires further verification.

### 2.2. Gp49 Does Not Affect the Growth of B. subtilis in Stationary Phase

A previous report suggested overexpressing plasmid-borne Gp49 in *B. subtilis* would not obviously affect the growth during the optimal growth condition [9]. As bacteria stay mostly in the stationary phase in the natural environment and complex phages like T4 carry abundant genes during infection to cope with different environments [17], it was necessary to evaluate the effect of Gp49 during the stationary phage since the genome of SPO1 is comparable to that of T4. We then repeated the growth attenuation experiment with an extended observation time. While the overexpressed Gp49 clearly did not affect the growth of *B. subtilis* during the early lag and exponential phases, it does not have a statistically significant attenuation effect on the growth of the bacteria during the stationary phase (Figure 2A). Inspired by another gene product—PEIP of the HTM—whose overexpression defies the rule of Gram staining by making *B. subtilis* stains resemble a Gram-negative bacterium [8], we then used the Gram stain for phenotyping the effect of overexpressing Gp49 during the stationary phase at OD_600_ = 1.4 (Figure 2B). Interestingly, the Gp49 overexpressing *B. subtilis* exhibited safranin color that characterizes Gram-negative bacteria, while bacteria in the pHT01 (control) groups showed the typical crystal violet for Gram-positive bacteria under the stain. As *gp49* is a nonessential gene for the phage, the growth attenuation assay and the Gram stains suggest Gp49 may have a role for the phage in invading *B. subtilis*.

### 2.3. Gp49 Has a Putative Nucleic Acid Binding Motif and Binds to Single-Stranded RNA

We thus attempted to elucidate the function of Gp49 by solving the structure of Gp49 using standard triple resonance nuclear magnetic resonance (NMR) spectroscopy with recombinantly expressed ^15^N and ^13^C labeled Gp49 (Figure 3A,B). The solution structure consists of six β strands and two helices, with the two helices on the two opposite sites of the β sheet (Figure 3C). Echoing its low similarity to the proteins with known structure, the homologue search using either Dali server [18] or PDBefold [19] suggests that Gp49 has a fold that was not described in the database, as no statistically significant similarity was found to other known proteins. However, the electrostatic surface revealed the existence of an accumulated positively charged patch formed by residues K35, K37, and R13 on one side and a negatively charged stripe on the other side of the protein (Figure 3D). Furthermore, the surface hydrophobicity of Gp49 suggests that the positively charged patch overlaps with the hydrophobic residues V36 and L67, forming a tandem hydrophobic–hydrophilic positively charged patch (Figure 3E). Notably, the surface exposed patch complements the negatively charged phosphate group in the ribonucleic acid (RNA) and deoxyribonucleic acid (DNA) and the hydrophobic nucleobase of single stranded nucleic acid. Thus, the structure suggests Gp49 might be a single-stranded (ss) nucleic acid binding protein.

To verify if Gp49 is indeed a nucleic acid binding protein, we performed electrophoretic mobility shift assays (EMSA) with agarose gel. Using the oligos listed in Appendix A, we performed EMSA assays using ssRNA, ssDNA, and dsDNA (Figure 4A–C). In the two Gp49-DNA EMSA assays, the migration of neither single-stranded or double-stranded DNA affected the addition of Gp49 (Figure 4A,B). However, Gp49 indeed interacts with ssRNA as the migration of RNA was affected by Gp49 at a dose dependent manner (Figure 4C). Using surface plasma resonance (SPR), we verified the interactions between Gp49 and ssRNA with a binding affinity of K_D_ = 1.86 ± 0.04 μM. In conclusion, Gp49 is a novel RNA binding protein, whose structure does not resemble any other published RNA binding protein.

### 2.4. Gp49 Upregulates Host Iron Metabolism

As Gp49 is an RNA binding protein and expressed during the early stage of phage development as *gp49* regulated by SPO1 early promoter [2], the host RNA should be its primary target. We then measured the transcriptome of Gp49 overexpressing *B. subtilis* using RNA-seq to check if the interaction between Gp49 and RNA affect the global transcription of the bacterium. The number of differentially expressed genes (DEGs) with a threshold of log2 fold change >1.0 and padj values ≤ 0.005 are 318, including 146 upregulated and 172 down regulated among 4900 total genes. According to the gene ontology (GO) categorization, most of the upregulated genes are involved in tetrapyrrole biosynthesis and metabolism in the biological process category, with heme and tetrapyrrole binding the molecular function category (Figure 5A). These upregulated genes include genes in *hem* operons and other genes that regulate heme synthesis and iron acquisition (Figure 5B and Appendix A). Similarly, porphyrin metabolism is the most upregulated process, in the Kyoto Encyclopaedia of Genes and Genomes (KEGG) analysis (Figure 5C). Thus, these results suggest Gp49 upregulates bacterial heme synthesis, an iron-containing porphyrin (tetrapyrrole), promoting the iron capture ability. Using quantitative PCR (qPCR), we verified the upregulation effect due to overexpression of Gp49 using *hemY* and *hemH* (Figure 5E). Meanwhile, the downregulation effect was not obvious in the Gp49 overexpression of *B. subtilis* (Figure 5D). This could be explained in that Gp49 does not have RNase activity and its RNA binding ability may stabilize, thus upregulate the genes in the *hem* operon. A recent study showed that the Gram-positive bacterium *Enterococcus faecalis* (*E. faecalis*) was stained safranin color instead of violet during Gram staining in the iron-supplemented condition (300 μM) [20]. The 11-fold increase of iron storage in bacterial cells reported in the study is consistent with our observation in the Gp49 overexpressing *B. subtilis*. As bacteriophage inactivation was shown to correlate with the concentration of ferrous iron or Fe (II) at low doses [14], producing Gp49 during the early stage of invasion to increase the capture of Fe (II) may provide survival advantage for SPO1 phage development.

## 3. Discussion

*B. subtilis* and its relatives have been considered gut commensals, with features being advantageous to survival within the human gastrointestinal tract (GIT), compared with soil strains [21]. *Bacillus*-based probiotics have been shown to improve digestive health by strengthening the intestinal barrier and limiting inflammatory responses [22]. Remodulated gut microbiome using phage has shown interesting results [23]. In addition to the phage-host relationship that is important to *B. subtilis* in the GIT, phage-mediated horizontal gene transfer (HGT) has profound consequences for human health and disease [24]. Thus, lytic bacillus phages like SPO1 have important roles in the homeostasis and dysbiosis of intestinal microbiota. As one of most studied phages infecting Gram-positive bacteria, SPO1 has been proven to be a rich source for discovering new strategies that the phage used to facilitate its progeny production. In particular, the HTM encodes gene products that shut off host DNA, RNA, and protein synthesis, and inhibit cell division by interacting with FtsL and HU [7,18,25]. As a novel RNA binding protein that regulates iron metabolism, Gp49 is another example of new phage strategy discovered by studying the HTM gene product.

Iron is an essential element for almost all living organisms, including bacteria. Bacteria have evolved sophisticated mechanisms to regulate iron uptake, storage, and utilization in order to maintain cellular homeostasis and ensure survival in various iron environments [11,26]. The regulation of iron in bacteria is also controlled at transcription level, through transcription factors such as Fur and non-coding regulatory RNAs like RyhB that interact with messenger RNAs (mRNAs) that are involved in iron metabolism [27]. Meanwhile, excess iron can be stored inside bacterioferritins to sequester iron, protecting the cell from oxidative stress [28]. In addition to iron utilization and storage, bacteria also use iron as a defense mechanism to counteract the host’s immune response and to compete with other microorganisms [29]. For example, bacteria can secrete siderophores into the host’s tissues, where these molecules chelate iron and form stable complexes with it. By sequestering iron, bacteria can deprive the host’s immune cells and compete with microorganisms, helping them to evade immune response and gain a competitive advantage [30].

While iron can also trigger prophage induction [31], phages are also susceptible to ferrous iron oxidation and can be inactivated when the environmental concentration of iron increases [14]. After entering the cell, phage starts taking over the host metabolism and reorientating the host machinery solely for its own replication. The iron balance may be disrupted if the host iron regulation becomes unavailable, which may also affect the development of the phage. By promoting the transcription form the host *hem* operon, SPO1 would gain advantage under certain external conditions, such as with an excess amount of environmental iron. Thus, our study revealed that Gp49 is a novel RNA binding protein that regulates host iron metabolism at transcriptional level. We propose that Gp49 would provide survival advantage for the phage while encountering excess iron from either cellular or external sources.

## 4. Material and Methods

### 4.1. Bacterial Growth Curve and Gram Staining

Gp49 was amplified from SPO1 genome by PCR using the primers listed in Appendix A, cloned into the pHT01 vector, and transformed into the *B. subtilis 168* strains. The cultures were then grown overnight at 37 °C in LB liquid medium supplemented with 25 μg/mL chloramphenicol. Then, 1% of the cultures were transferred to 2 × YT medium containing 2% (*w*/*v*) glucose (to prevent leakage of pHT01 vector), 25 μg/mL chloramphenicol, and either water or 1mM IPTG. Cultures were conducted in 96-well plates (Corning, Corning, NY, USA) with a volume of 200 μL per well. The 96-well plates were assessed for OD_600_ using a Cytation 5M enzyme marker (Bio-Tek, Winooski, VT, USA). The cultures were incubated at 37 °C with continuous shaking at 573 rpm for a duration of 12 h. Three biological and technical replicates were performed.

At the end of the 12 h growth curve, 10 μL of each bacterial solution from the four groups was absorbed onto a slide and rapidly fixed at high temperature. Following primary staining, mordanting, decolorization, and counterstaining, bacterial morphology was observed under an oil immersion microscope at 200× magnification (Zeiss Axio scopeA1, Zeiss, Oberkochen, Germany).

### 4.2. Protein Expression and Purification

SPO1 gene 49 was amplified by PCR and ligated into pET-28a expression vector. The plasmid was transformed into BL21 (DE3) cell strain, and the cells were grown at 37 °C to OD_600_ of 0.6–0.8 and induced with 1 mM of isopropyl-b-D-thiogalactopyranoside (IPTG) in LB media. Soluble Gp49 was purified by immobilized metal-ion chromatography with a Ni-NTA column. After purification, the C-terminal 6xHis-tagged Gp49 was immediately dialyzed overnight at 4 °C into the buffer (250 mM NaCl, 50 mM Na_2_HPO_4,_ and pH 7.5). For NMR study, the ^15^N and ^13^C labelling of Gp49 was achieved using M9 minimal medium containing 0.07% ^15^NH_4_Cl and 0.2% ^13^C-glucose (CIL).

### 4.3. Structure Determination

The structural determination for Gp49 was performed in the buffer described above with a final concentration of 0.7 mM. The spectra were collected at 298 K on a Bruker 800 MHz (Avance NEO), Billerica, MA, USA, spectrometer equipped with cryo-probe. The backbone assignment was completed using our in-house, semi-automated assignment algorithms and standard triple-resonance assignment methodology using HNCACB/CBCANH, and HNCO/HN(CA)CO. H_α_ and H_β_ assignments were obtained using HBHA(CBCACO)NH and the full side-chain assignments were extended using HCCH-total correlation (TOCSY) spectroscopy and (H)CCHNH-TOCSY. Three-dimensional ^1^H-^15^N/^13^C NOESY-HSQC experiments provided the distance restraints used in the final structure calculation. Three-dimensional ^1^H-^15^N/^13^C NOESY-HSQC (mixing time 100 ms at 800 MHz) experiments provided the distance restraints used in the final structure calculation. The ARIA protocol was used for completion of the NOE assignment and structure calculation. The structural statistics are shown in Appendix A.

### 4.4. RNA Synthesis

The RNA was synthesized in vitro using RiboMAX Large Scale RNA Production System (Promega, Madison, WI, USA). The template DNA containing 21-nt complementary strands were obtained by mixing equivalent molar amounts of each oligonucleotide, then adding the enzyme mix, RiboMAX™ Express T7 2X Buffer, linear DNA template (1 µg total), nuclease-free water, mixing gently, and incubating at 37 °C for 3 h. The RNA was then purified using gel extraction purification.

### 4.5. Electrophoretic Mobility Shift Assay (EMSA)

These assays used RNA, ssDNA, and dsDNA fragments (the sequence is listed in Appendix A) as substrates in a buffer consisting of 20 mM Tris and 50 mM NaCl (pH 7.5). RNA oligonucleotides (RNA, 5′-AUGAAUAGUUAAACAACGUGG) were obtained from T7 RiboMAX™ Express Large Scale RNA Production System. The reaction mixtures, containing different reagents as illustrated in Figure 4, were topped up with the buffer to give a final volume of 6 μL and incubated for 30 min at 4  °C. The total binding solutions were loaded onto 2% agarose gels in 0.5× TBE buffer. All EMSAs were performed in duplicate, and the results show representative gels.

### 4.6. Surface Plasmon Resonance

For the analysis using SPR, Gp49 protein (10 μg/mL) was fixed on the COOH sensor chip (CM5) by capture-coupling, the interactions of fixed Gp49 with ssRNA detected by Biacore T100 (GE Healthcare, Chicago, IL, USA) at 25 °C. The binding time was ~85 s and the disassociation time was approximately 120 s. The flow rate was 30 μL/s. A 1:1 diffusion corrected model was fitted to the wavelength shifts corresponding to the varied protein and ligand concentration. The data was retrieved and analyzed with Biacore T100 Evaluation Software.

### 4.7. RNA Extraction and RNA-Seq Analysis

*B. subtilis* (stain 168) containing empty pHT01 and pHT01-gp49 were induced with 0.5 mM IPTG for 4 h at 37 °C at OD_600_ = 0.6. Then, 1 mL cell culture was taken from each sample for RNA extraction. The bacteria RNA was extracted from uninduced and induced samples by RNA Extraction Kit (ThermoFisher, Waltham, MA, USA). RNA quality and quantity were checked using Bioanalyzer (Agilent, Santa Clara, CA, USA) and RNA 6000 Nano kit (Agilent, USA). For RNA-seq experiments, rRNA was removed using the Ribo-Zero rRNA removal kit. Tophat2 and Cufflinks 2.2.1 were used to analyze the RNA-seq data and identify differential genes [32]. The bacterial gene expression level was measured by FPKM (reads per kilobase per million read), and genes with log2 fold change values of 1.5 and q values of <0.05 were defined as differentially expressed genes (DEGs).

### 4.8. Real-Time Quantitative PCR (RT-qPCR)

RT-qPCR used *B. subtilis* 16S ribosomal RNA as the internal control. Total RNA was extracted from each sample using RNA fast200 Kit (FASTAGEN, Shanghai, China, Cat#220010 50), according to the manufacture’s instruction. Then, 5 μg of total RNA was reverse transcribed with qPCR RT Master Mix Kit (TOYOBO, Osaka, Japan, Cat#FSQ-301) for complementary DNA synthesis. RT- qPCR was performed in a 20 μL volume consisting of 10 μL 2X QuantiNova SYBR Green PCR Kit (QIAGEN, Hilden, Germany, Cat#208054), 8 μL RNA-free water, 1 μL cDNA template, and 0.5 μL of each primer pair according to the manufacturer’s procedure using CFX Connect Real-Time PCR System (BIO-RAD, Hercules, CA, USA). Three biological replicates were performed each with three technical replications. Relative differential expression levels of *msyB* were calculated based on the cycle threshold (C_T_) values normalized with the control gene using the 2^−△△CT^ method.

## Figures and Tables

**Figure 1 ijms-24-14318-f001:**
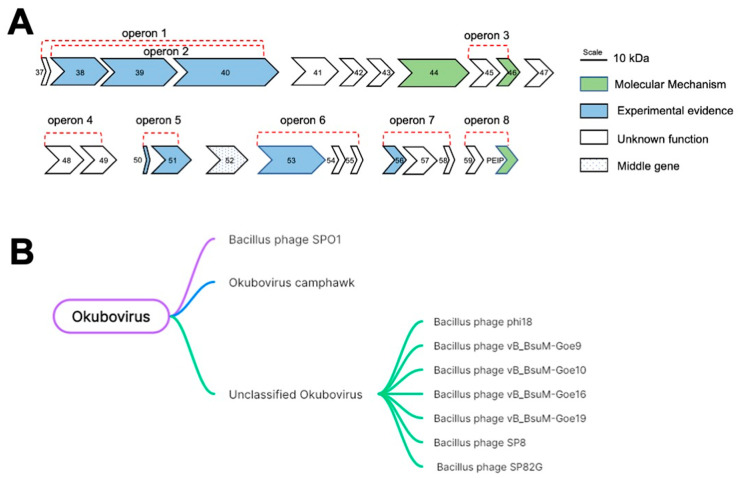
The location of Gp49 in the HTM and its conversation among phages. (**A**) The current understanding of each gene in the HTM as illustrated. Each gene product is scaled to length according to its molecular weight. (**B**) The taxonomy tree showing the conservation of Gp49 among *Okubovirus* using BLAST.

**Figure 2 ijms-24-14318-f002:**
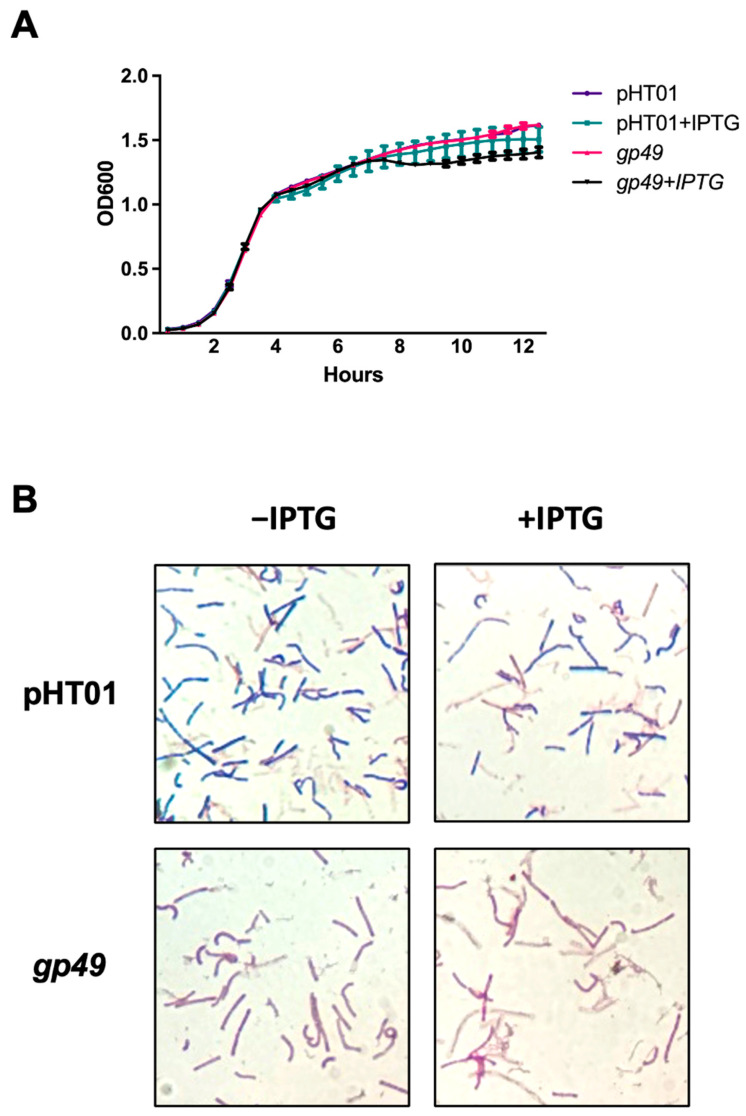
The phenotypes caused by overexpressing Gp49. (**A**) The growth curves of *B. subtilis* cells bearing empty pHT01 plasmid and *B. subtilis* cells bearing pHT01-*gp49* plasmid before and after IPTG induction. Three biological and technical replicates were performed for each growth curve. (**B**) Gram staining images of control *B. subtilis* cells bearing empty pHT01 plasmid and *B. subtilis* cells bearing pHT01-*gp49* plasmid before and after IPTG induction. Three biological replicates were performed for the Gram staining.

**Figure 3 ijms-24-14318-f003:**
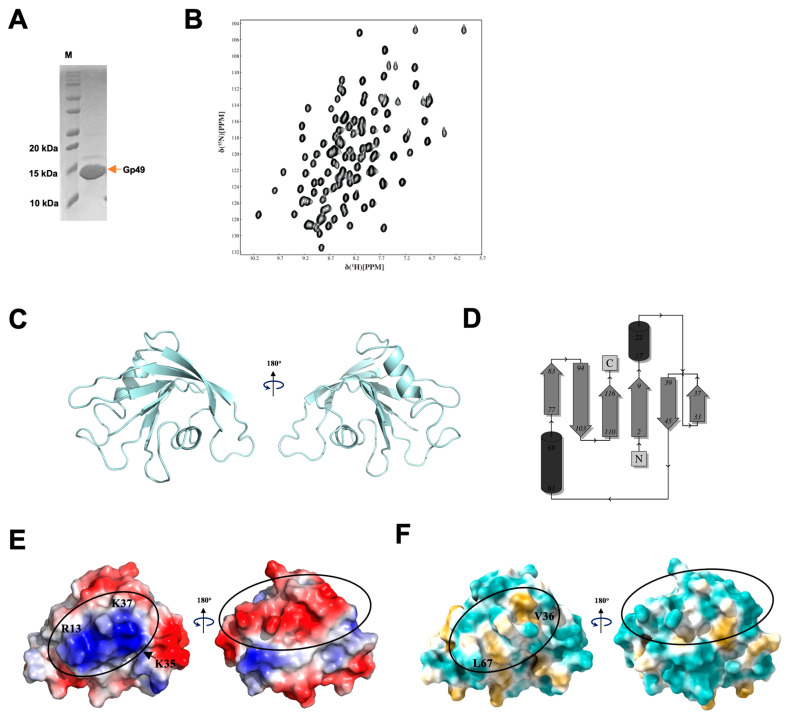
The solution structure of Gp49. (**A**) The SDS-PAGE image showing the recombination expression of Gp49. (**B**) The ^1^H-^15^N HSQC spectrum of ^15^N-labelled Gp49. (**C**) Cartoon representations of Gp49 shown in 180° rotation. (**D**) The topology of Gp49 generated by PDBsum. (**E**) The electrostatic surface of Gp49 as shown in (**A**) in 180° rotation. (**F**) The surface hydrophobicity of Gp49 as shown in (**A**) in 180° rotation (blue: hydrophilic, white: neutral and yellow: hydrophilic).

**Figure 4 ijms-24-14318-f004:**
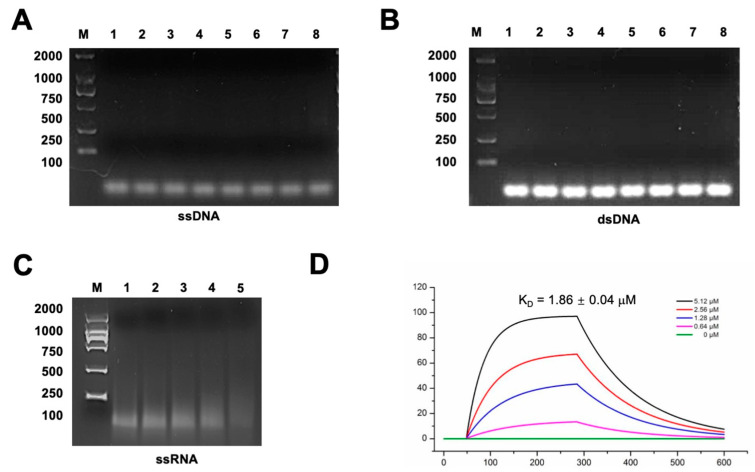
Gp49 interacts with ssRNA. (**A**) The image of agarose gel showing EMSA assay for Gp49 and ssDNA. From lane 1 to lane 8, the ratios of ssDNA versus Gp49 are 1:0, 1:1, 1:2, 1:4, 1:8, 1:16, 1:32, and 1:100, respectively. (**B**) The agarose gel image showing EMSA assay for Gp49 and dsDNA. From lane 1 to lane 8, the ratios of ssDNA versus Gp49 are 1:0, 1:1, 1:2, 1:4, 1:8, 1:16, 1:32, and 1:100, respectively. (**C**) The agarose gel image showing EMSA assay for Gp49 and ssRNA. From lane 1 to lane 5, the ratios of ssDNA versus Gp49 are 1:0, 1:0.25, 1:0.5, 1:1, and 1:2, respectively. The sequences of the nucleic acids are listed in Appendix A. The nucleic acids used in the EMSA experiments were all at 1 μM. Three biological and technical replicates were performed for EMSA assays, and the representative images are shown. (**D**) The affinity for ssRNA and Gp49 interaction measured by SPR.

**Figure 5 ijms-24-14318-f005:**
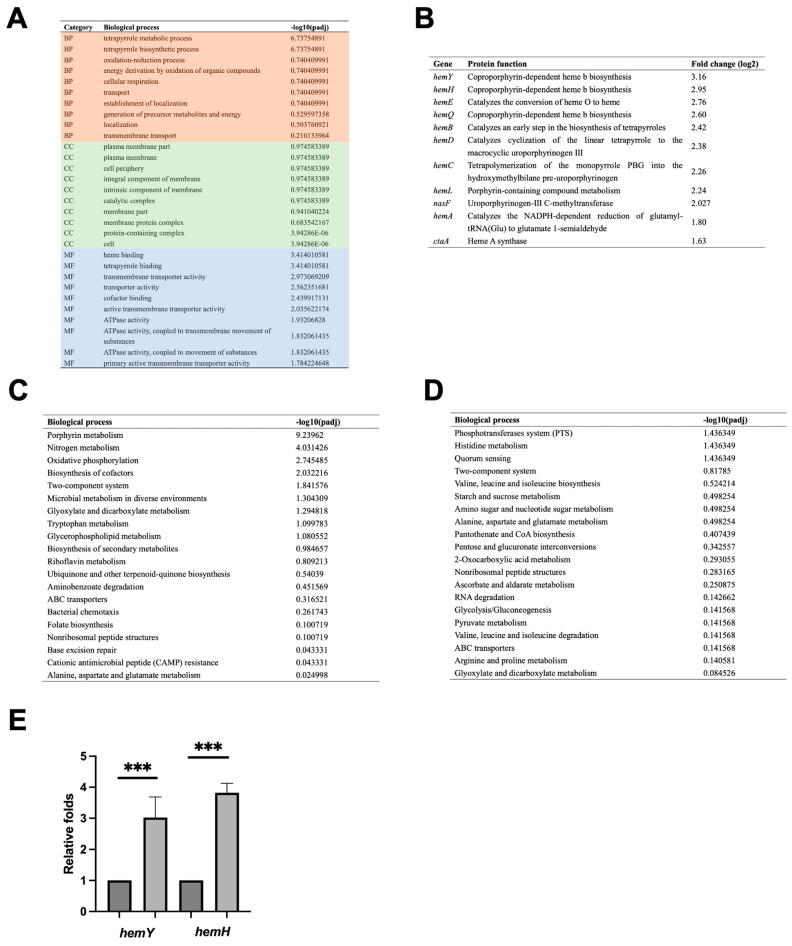
Transcriptome analysis of Gp49 overexpression. (**A**) The most upregulated genes category according to GO analysis. BP: Biological Process, MF: Molecular Function and CC: Cellular Component. (**B**) The upregulated genes that are involved in tetrapyrrole metabolism and biosynthesis as demonstrated in (**A**). (**C**) The most upregulated biological process according to KEGG analysis. (**D**) The most downregulated biological process according to KEGG analysis. (**E**) The level of *hemY* and *hemH* mRNA fold changes before and after IPTG induction. Three biological and three technical replicates were performed for the RNA-seq and qPCR. *** *p* < 0.001.

## Data Availability

The NMR structures of Gp49 were deposited in the wwPDB OneDep System with a PDB ID: 8K58. The transcriptomic sequencing data generated from this study were deposited in the NCBI Sequence Read Archive (SRA) under BioProject accession no. PRJNA1008287.

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
