# Peer review of "Phage SPO1 Protein Gp49 Is a Novel RNA Binding Protein That Is Involved in Host Iron Metabolism"

_ijms, 2023, doi:10.3390/ijms241814318_

Round 1
Reviewer 1 Report

Please check the text carefully for syntax and typing errors (sometimes words are missing...).
Author Response
Major comments
- Although a good functional characterization has been done, the paragraphs relating to NMR spectroscopy, both in the results and in the materials and methods, is severely limited and poor in data. The authors should clarify what is the concentration of the protein used for the NMR experiments, which experiments were acquired and which softwares were used for data analysis.
We thank the reviewer for point this out. We have now added revelant description for this NMR experiments.
- Furthermore, it would be interesting to insert in the main text a figure that shows an NMR spectrum, even just an (1H-15N) HSQC.
We thank the reviewer for point this out. We have now added the HSQC spectrum in Fig. 3B and modified the relevant text.
- In the main text, page 9, paragraph 4.5, the authors refer to a table 1 which I have not found.
We thank the reviewer for pointing this error. The correction has been made.
- In table S3, where "Average Pairwise rmsdα(AÌŠ)" is reported, what does the superscript "a" refer to?
We thank the reviewer for pointing this mistake out. The correction has been made.
Minor comments
- Thefigure 5 is notclear and InparticularI suggest making a table regarding figure 5B.
- Ifoundsometypographical/unintentionalerrorsmentionedbelow:
- pag. 5, line 6: “K37” is written twice. I believe that it should be replaced with K35 as well as shown in the 3D figure .
- pag. 5, line 8: "Futhermore" should be written with a capital F.
We thank the reviewer for valuable comments. We have remade the Fig. 5 to improve the readability and corrected the errors accordingly.

Reviewer 2 Report
The manuscript by Yang Y. et al. entitled "Phage SPO1 protein Gp49 is a novel RNA binding protein that involves in host iron metabolism" reports the characterization of Gp49 protein as a transcription regulator that promotes iron metabolism at transcriptional level. The authors show that Gp49 folds into a structure that does not resemble other published protein and contains a putative RNA binding motif.
Although the manuscript is interesting, I suggest to publish it after major revisions.
Major comments
-
Although a good functional characterization has been done, the paragraphs relating to NMR spectroscopy, both in the results and in the materials and methods, is severely limited and poor in data. The authors should clarify what is the concentration of the protein used for the NMR experiments, which experiments were acquired and which softwares were used for data analysis.
-
Furthermore, it would be interesting to insert in the main text a figure that shows an NMR spectrum, even just an (1H-15N) HSQC.
-
In the main text, page 9, paragraph 4.5, the authors refer to a table 1 which I have not found.
-
In table S3, where "Average Pairwise rmsdα(Å)" is reported, what does the superscript "a" refer to?
Minor comments
-
The figure 5 is not clear and should be improved in quality. In particular I suggest making a table regarding figure 5B.
-
I found some typographical/unintentional errors mentioned below:
- pag. 5, line 6: “K37” is written twice. I believe that it should be replaced with K35 as well as shown in the 3D figure .
- pag. 5, line 8: "Futhermore" should be written with a capital F.

Author Response
Results 2.1- Analysis of the gene's genetic context (function of other genes close to it) suggests that the protein may be involved in host nucleic acid metabolism, but gives little information.
We thank the reviewer for valuable comment. We have modified the statements to improve the readily and the flow the information.
Results 2.2- They then confirm that over-expression of Gp49 does not affect the growth of B. subtilis. They then use Gram labeling to go further (Figure 2B).... but this part of the work is incomprehensible to me. I understood neither the experiment nor the conclusion drawn from it... I would appreciate it if the authors would rewrite this paragraph so that the experiment is clearly explained and the conclusions understandable 2.2.
We thank the reviewer for valuable comments. We have modified the statement and improved the logic for performing the two unconnected experiments.
Results 2.3- The sequence of the protein was too divergent from the proteins listed in the databases (it would have been useful to have this sequence in the article), so the authors solved the 3D structure of Gp49 in an attempt to discover its function by structural analogy. Unfortunately, the resulting structure is new... I would have liked to have the coordinates to be able to check for myself. Analysis of the structure shows the presence of a positively charged patch which may be involved in interaction with nucleic acids. This hypothesis has been successfully tested in EMSA and SPR, showing specificity for ssRNA.
We thank the reviewer for pointing this out. The sequence of Gp49 and a screenshot of the BLAST results are listed below for your convenience.
MIKAAVTKESLYRMNTLMEAFQGFLGLDLGEFTFKVKPGVFLLTDVKSYLIGDKYDDAFNALIDFVLRNDRDAVEGTETDVSIRLGLSPSDMVVKRQDKTFTFTHGDLEFEVHWINL
Results 2.4 - The first sentence of this paragraph "expressed during early stage of phage development" makes me wonder: how do the authors know this? Perhaps it has something to do with Figure 2B, which I didn't understand? The transcriptomics experiment is interesting though, and demonstrates unambiguously (although I would have appreciated access to tables S1 and S2 to be sure) that Gp49 is probably involved in iron metabolism in B. subtilis.
We thank the reviewer for pointing this out. We have added reference for the statement as Gp49 has an early promoter.
The discussion is interesting and puts this work into perspective for the use of B. subtilis.
Minor :
Introduction : ...encoding proteins that ARE essential...
2.3 : ...The solution structure consists of six β STRANDS... ... two opposite site of the β SHEET... 2.3 : ... formed by residues K35, K37 and R13...
2.3:... Furthermore (capital letter)
Materiel and Methods :
4.2 : please add Gp49 labeling protocol for NMR experiments 4.5 : as illustrated in Fig. 4 instead of 1
We thank the reviewer for pointing the mistakes. We have now corrected the errors accordingly.

Round 2
Reviewer 2 Report
The authors answered all doubts.
I would recommend publishing this paper.